# You Always Recognize Me (YARM): Robust Texture Synthesis Against Multi-View Corruption

**Weihang Ran** [1]  **Wei Yuan** [2]  **Yinqiang Zheng** [1]

## Abstract

Damage to imaging systems and complex external environments often introduce corruptions, which can impair the performance of deep learning models pretrained on high-quality image data. Previous methods have focused on restoring degraded images or fine-tuning models to adapt to out-of-distribution data. However, these approaches struggle with complex, unknown corruptions and often reduce model accuracy on high-quality data. Inspired by the use of warning colors and camouflage in the real world, we propose designing a robust appearance that can enhance model recognition of low-quality image data. Furthermore, we demonstrate that certain universal features in radiance fields can be applied across objects of the same class with different geometries. We also examine the impact of different proxy models on the transferability of robust appearances. Extensive experiments demonstrate the effectiveness of our proposed method, which outperforms existing image restoration and model fine-tuning approaches across different experimental settings, and retains effectiveness when transferred to models with different architectures. Code will be available at https://github.com/SilverRAN/YARM.

## 1. Introduction

Neural network-based deep learning technologies have made significant impacts across various domains in modern society, including facial recognition, autonomous driving, and 3D reconstruction. However, previous research has shown that neural network models can yield erroneous predictions in certain scenarios, such as under adversarial attacks or image degradation (Hosseini et al., 2017; Dodge

[1]The University of Tokyo, Japan [2]Tohoku University, Japan. Correspondence to: Yinqiang Zheng <yqzheng@ai.u-tokyo.ac.jp>.

*Proceedings of the 42nd International Conference on Machine Learning*, Vancouver, Canada. PMLR 267, 2025. Copyright 2025 by the author(s).

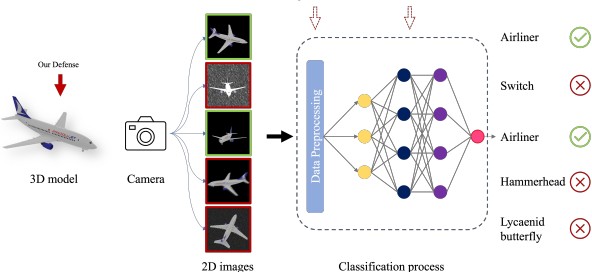

*Figure 1.* The proposed method is illustrated in this diagram. During imaging of a real-world object, various types of degradation (e.g., noise from damaged camera components, blur from object motion, or adverse weather conditions like snowfall) can be introduced, resulting in low-quality images that degrade performance in downstream tasks (e.g., image classification). Previous approaches have primarily focused on data preprocessing (image restoration) or model fine-tuning. In contrast, we propose addressing this issue from a data perspective, enhancing robustness to low-quality imaging by altering the appearance of natural objects.

& Karam, 2017; Geirhos et al., 2017). For instance, commonly encountered conditions like rain or snow can reduce the accuracy of target detection models in recognizing traffic signs (Wang et al., 2022). This lack of robustness hinders the advancement of neural network technologies in industrial applications where high accuracy and reliability are essential. Addressing these issues is also critical for the transition from Artificial Narrow Intelligence (ANI) to Artificial General Intelligence (AGI).

Previous studies have proposed two primary approaches to address this issue (see Fig.1): input preprocessing (Pei et al., 2018; Liu et al., 2018; Son et al., 2020), and model fine-tuning (Wang et al., 2020; Kim et al., 2021; Yang et al., 2023). Since model accuracy often degrades due to input data being attacked or corrupted, restoring low-quality data to its original state could theoretically prevent prediction errors. This approach typically employs image restoration techniques to remove corruptions in the input image, aiming to recover its original content as closely as possible. Methods include image denoising, deblurring, rain and fog removal, and super-resolution. However, these techniques

generally focus on the quality of the image restoration rather than the restored image's effectiveness in downstream tasks. On the other hand, some researchers argue that changes in input images are typically insufficient to impair human judgment, suggesting that model performance declines because neural networks lack the robustness of human perception. By applying adversarial training or fine-tuning, it is possible to improve model performance on corrupted data. However, previous studies have shown that the effectiveness of this approach remains limited.

To address this issue, we propose an innovative approach. In the real world, it is well known that the ease with which objects are perceived by humans can be modified by changing their colors and textures. For instance, traffic cones are typically painted bright red, while military vehicles like tanks are designed with camouflage. Similarly, could we synthesize a texture for object surfaces that makes them easier for deep learning models to recognize, regardless of the environment? Unlike previous methods, our study focuses on mitigating the effects of data degradation on model performance from a data-centric perspective. Consequently, in industrial applications, manufacturers could leverage our research to design product appearances that enhance recognizability. For example, in a future where autonomous driving systems based on computer vision are widely deployed, a bicycle designed with textures that enable accurate identification under various conditions would be preferable to a standard bicycle, as it could help reduce accident risks. Based on this rationale, we believe that dual enhancements in both data and model design are essential for developing highly reliable AI systems, which imparts significant societal relevance to our work.

In this paper, we investigate three specific questions: (1) Can texture optimization enhance neural network models' object recognition performance, achieving better results than previous methods? (2) Is it possible for these robust textures to be transferable, allowing them to generalize across objects with different geometries? (3) Do robust textures generalize effectively across different neural network architectures? To address these questions, we propose a method for synthesizing robust textures in this paper. For object-specific robust texture optimization, we first obtain a voxel representation containing the object's geometry and color by employing 3D reconstruction on multi-view images. Next, we randomly select a viewing angle, render the corresponding 2D image, and apply various image degradation operations of differing intensities, including noise, blur, weather effects, and compression. A classifier is used as a surrogate model to recognize the degraded images, and backpropagation is performed to optimize the color features of the voxel representation. For the optimization of universal robust textures, we first reconstruct voxel representations for multiple objects of the same category based on multi-view image sets.

We then initialize a random perturbation delta, applying it to the features of different voxel representations for optimization. Extensive experiments demonstrate that these robust textures show significantly improved resistance to image corruptions compared to previous methods and can be optimized to produce a universal texture adaptable to objects of the same category with varying geometries.

The primary contributions of our research are as follows:

- We propose a data-centric approach to enhance the performance of deep learning models in the presence of image corruptions. By using multi-view 3D reconstruction to obtain a voxel representation of the target object and optimizing for a robust texture, our method significantly improves model recognition accuracy without requiring any preprocessing or model fine-tuning.

- We demonstrate the existence of a universal robust texture that can transfer across objects of the same category but with different geometries. This universal texture effectively aids downstream models in resisting degraded imaging, even in zero-shot scenarios.

- Through an analysis of the transfer performance of robust textures generated under various surrogate models, we establish insights into how different models impact final performance. Based on this, we propose a method for selecting the most suitable surrogate model.

## 2. Related Works

### 2.1. Visual Recognition against Image Corruptions

Image corruption during the imaging process often leads to suboptimal performance in downstream deep learning models (Hosseini et al., 2017; Dodge & Karam, 2017; Geirhos et al., 2017). Therefore, how to improve accuracy on corrupted input images is an urgent problem to address. (Hendrycks & Dietterich, 2018) firstly introduced an image dataset containing 15 types of corruptions and evaluated the robustness of various deep learning models against different kinds and severities of corruption. (Bai et al., 2021) compared the performance of CNNs and Transformers on corrupted images under a fairer setup, revealing the reasons why Transformer architectures demonstrate superior generalization on out-of-distribution (OOD) data.

To reduce model vulnerability to low-quality images, some studies have attempted to fine-tune models for different kinds of corruption. However, (Vasiljevic et al., 2016; Zheng et al., 2016) found that models fine-tuned on a single type of blur did not generalize well to other types, while fine-tuning on multiple degradation types often resulted in decreased overall performance. Works such as (Wang et al., 2020; Kim et al., 2021) have shown that model performance decline

is due to the degradation of the deep feature representation space. These approaches aim to map degraded features to clean features to improve accuracy in downstream tasks. (Yang et al., 2023) leverages vector quantization to bridge the gap between low-quality and high-quality features, learning representations that are invariant to quality. Additionally, (Liu et al., 2024) identified that the channel correlation matrix of features is a reliable indicator of degradation type and provides a clear optimization direction for unsupervised solution space by reducing the difference between the channel correlation matrices of degraded and clean features.

Additionally, some studies have attempted to restore degraded images directly to preserve the performance of downstream models. Although there is already extensive research on image restoration (Zhang et al., 2021; Ren et al., 2019; Ji et al., 2023), (Pei et al., 2018) noted that simply applying dehazing operations to images does not improve accuracy in downstream classification tasks, as restored images still differ from high-quality images in feature space. To address this, (Liu et al., 2018; Son et al., 2020) have jointly optimized the image restoration module alongside high-level models to make the restored images more suitable for downstream tasks, finding that this approach also enhances restoration effectiveness.

Apart from these two methods, some research has explored robustness in 3D objects, such as (Salman et al., 2021; Wang et al., 2022; Lin et al., 2025). However, these approaches have not been thoroughly tested on large-scale datasets or across different model architectures, nor do they offer a straightforward, user-friendly workflow.

### 2.2. NeRF-based 3D Editing

Novel view synthesis is a long-standing research topic in the field of 3D reconstruction. Its goal is to synthesize images from previously unseen viewpoints, given a set of images that capture a scene. Traditional approaches include direct interpolation across densely captured scenes and combining depth maps to handle sparse viewpoints (Buehler et al., 2001; Shi et al., 2014; Shih et al., 2020). However, these methods often come with significant limitations. With the advancement of neural rendering, novel view synthesis methods based on neural radiance fields (NeRF) (Mildenhall et al., 2020; Barron et al., 2021; 2022) have shown immense potential. NeRF employs a multi-layer perceptron (MLP) with positional encoding as an implicit and continuous volumetric representation. Impressive visual results and flexible configuration make NeRF a suitable foundation for further 3D editing tasks, including geometric transformations (Yuan et al., 2022; Yang et al., 2022), style transfer (Wang et al., 2023; Zhang et al., 2022), and text-to-texture synthesis (Richardson et al., 2023; Dong & Wang,

2024). Despite various acceleration techniques (Fridovich-Keil et al., 2022; Müller et al., 2022), NeRF's lengthy training times and slow inference speeds remain significant drawbacks. To address this, (Sun et al., 2022; Karnewar et al., 2022) have combined NeRF's original setup with explicit volumetric grid modeling, significantly accelerating both training and inference. Subsequent work (Sella et al., 2023) has also shown that voxel grid-based 3D representations can be effectively used for scene editing.

## 3. Methodology

### 3.1. Problem Formulation

Given a 3D object $X$ with a true class label $y$, when an imaging system captures a 2D image $v_i$ from a given viewpoint $i$ and applies it to a downstream task (e.g., classification), the downstream model $f_\theta$ should provide the correct prediction theoretically, such that $f_\theta(v_i) = y$. However, due to complexities in the imaging process or environment, unknown corruption $C$ can be introduced. When a degraded image $v_i' = C(v_i)$ is used by the downstream classifier, this may lead to incorrect predictions, i.e., $f_\theta(v_i') = y' \neq y$.

The objective of our research is to optimize the appearance of the given 3D object $X$ to get a robust version $X^R$, which can ensure that its visual features remain robust against various types of corruptions during the imaging process, thereby improving the prediction accuracy of downstream models.

Furthermore, inspired by studies on universal adversarial perturbations (UAP) (Moosavi-Dezfooli et al., 2017; Hendrik Metzen et al., 2017), we explore the potential existence of a universal robust texture (URT). Specifically, given a set $\mathcal{X} = \{X_1\langle G_1, T_1\rangle, X_2\langle G_2, T_2\rangle, ..., X_i\langle G_i, T_i\rangle\}$ containing multiple objects of the same class $y$, where $G_i$ and $T_i$ represent the geometry and texture of object $X_i$, respectively. We aim to optimize a universal texture $T_U$ that can transform any object in the set into a robust version: $X_i\langle G_i, T_U\rangle \rightarrow X_i^R$.

### 3.2. 3D Reconstruction based on Voxel Grid

Given a set of multi-view images captured in a static scene, NeRF (Neural Radiance Fields) learns a mapping between each image's viewpoint coordinates and direction to the corresponding pixel values, constructing a neural radiance field $F(x, d) \rightarrow (c, \sigma)$. Here, the input $x \in \mathbb{R}^3$ represents coordinates within the radiance field, $d$ is the unit-norm viewing direction, and the output consists of $\sigma \in \mathbb{R}^+$ (the volume density) and $c \in [0, 1]^3$ (the emitted RGB color). During inference, given any arbitrary camera viewpoint, NeRF queries multiple sample points along rays emitted from the camera center and calculates pixel values at specific locations based on the volumetric rendering formula.

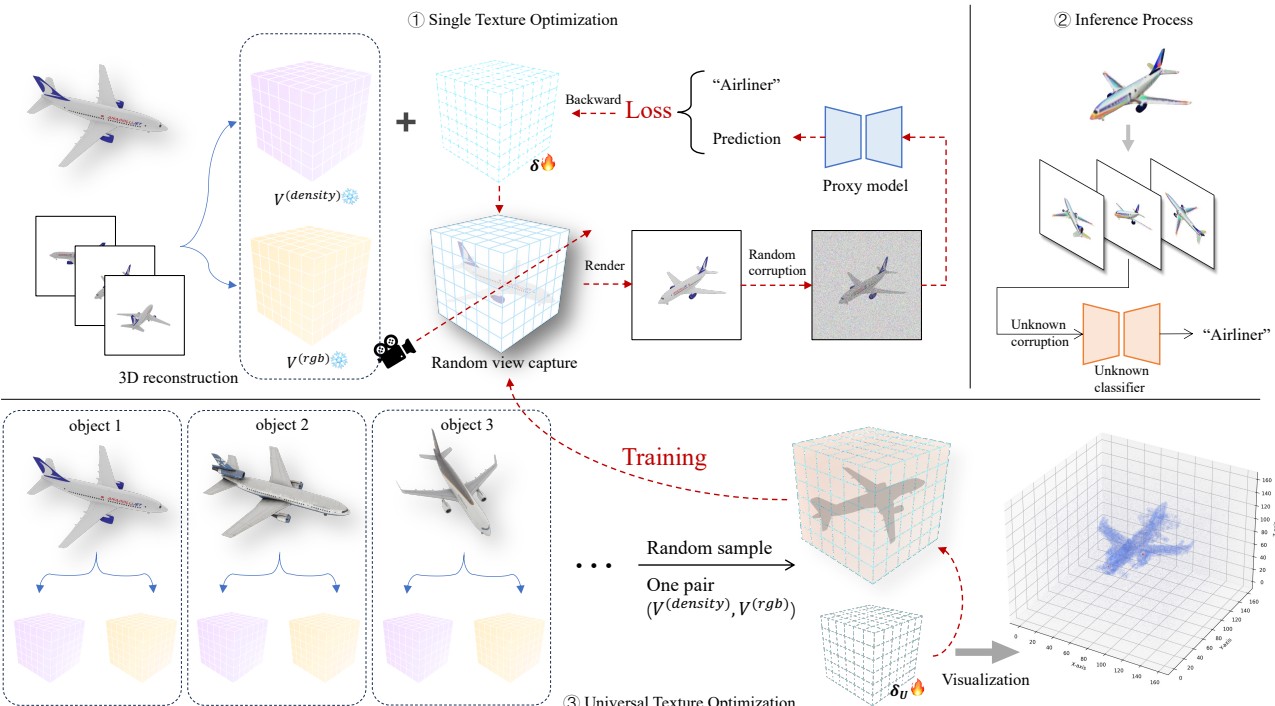

*Figure 2.* The framework of our proposed method is illustrated as follows. Given an object, we first collect multi-view images to reconstruct a 3D voxel representation. We then initialize a perturbation $\delta$, with the same shape as the color voxel grid and combine it with the original voxel representation to perform random-view rendering, producing 2D images. These images are then degraded by randomly selected types and intensities of corruption, and the degraded image is fed into a surrogate classifier to obtain predictions and compute loss for updating $\delta$'s parameters. For universal robustness textures, we first select multiple objects of the same category, reconstructing individual voxel representations for each. During training, in each iteration, a random pair of density and color voxel grids is selected, and a universal perturbation $\delta_U$ is applied, followed by the same training process described above. After training, the optimized $\delta_U$ can be combined with any reconstructed voxel representation to render robust images.

Although NeRF can establish a continuous and implicit radiance field, previous work has highlighted that editing its parameter space is challenging, and the training process is time-intensive. Consequently, our work employs an improved NeRF approach based on a voxel grid representation. Voxel grid representation explicitly models the scene's modalities of interest (e.g., density, color, features) using grid cells, offering higher query efficiency. For a point of interest $x$, its value within the voxel grid can be obtained through trilinear interpolation.

$$\text{interp}(\boldsymbol{x}, \boldsymbol{V}) : (\mathbb{R}^3, \mathbb{R}^{C \times N_x \times N_y \times N_z}) \rightarrow \mathbb{R}^C \quad (1)$$

where $x$ is the queried 3D point, $V$ represents the voxel grid, $C$ is the modality dimension, and $N_x \times N_y \times N_z$ is the total number of voxels. To reduce training difficulty, DVGO (Sun et al., 2022) employs a coarse-to-fine staged training approach. In the coarse stage, the target radiance field is initialized over a large spatial region, learning view-invariant colors $V^{(\text{rgb})(\text{c})} \in \mathbb{R}^{3 \times N_x^{(c)} \times N_y^{(c)} \times N_z^{(c)}}$ and raw volume densities $V^{(\text{density})(\text{c})} \in \mathbb{R}^{1 \times N_x^{(c)} \times N_y^{(c)} \times N_z^{(c)}}$. In the fine stage,

to capture finer surface details and view-dependent colors, the bounding box is progressively scaled down, and free space is skipped to accelerate query speeds. We apply an improved version (Karnewar et al., 2022) of this approach, replacing the softplus activation function with ReLU because it can preserve the discontinuities present in real-world signals. After training we can get two voxel grids $V^{(\text{density})(\text{f})}$ and $V^{(\text{rgb})(\text{f})}$.

### 3.3. Optimize Single Robust Texture

After obtaining the fine-stage voxel grids $V^{(\text{density})(\text{f})}$ and $V^{(\text{rgb})(\text{f})}$, we can optimize them to enhance robustness against image corruptions. We consider changing the object's geometry impractical in real-world scenarios, as the functionality of various items is closely tied to their shape. Therefore, we fix the density grid $V^{(\text{density})(\text{f})}$ obtained from the previous stage and only optimize the color grid. Adopting a setup similar to adversarial attacks, we initialize a perturbation $\delta$ with the same shape as the color grid and add

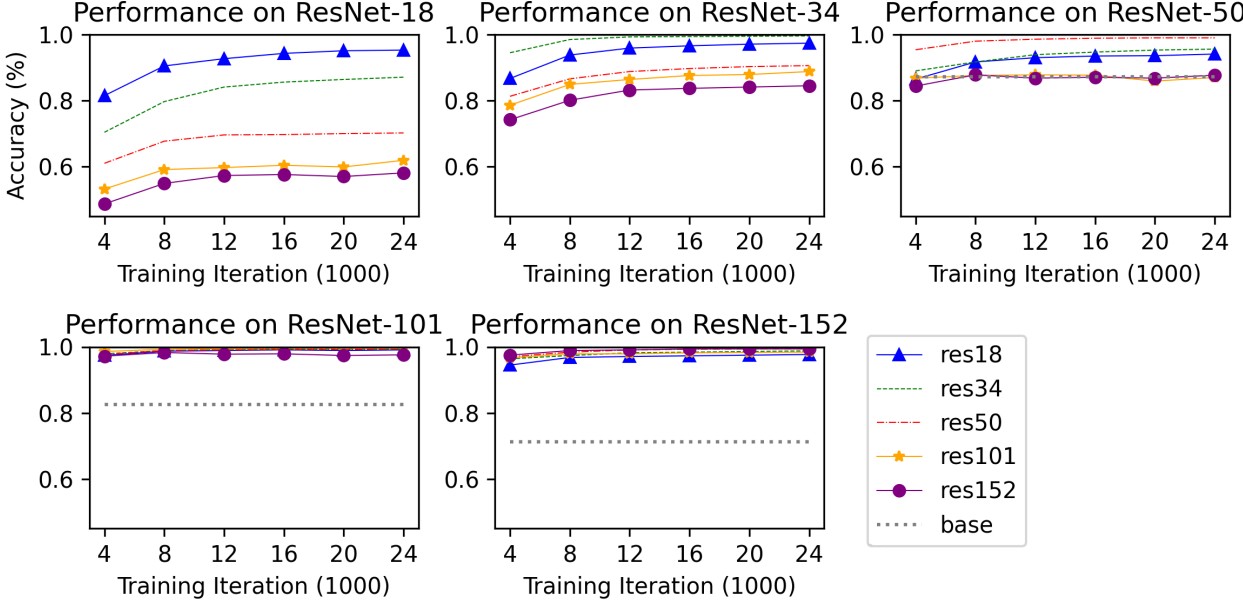

*Figure 3.* Transferable performance of textures on proxy model to other models. We used ResNet-18, ResNet-34, ResNet-50, ResNet-101, and ResNet-152 as surrogate models and transferred the resulting textures to other models for testing. It was observed that using a model as its own surrogate consistently yielded the best performance. Aside from this self-surrogacy, textures generated with smaller-parameter models as surrogates tended to perform better across other models, especially evident with ResNet-18, ResNet-34, and ResNet-50. In contrast, for ResNet-101 and ResNet-152, the performance differences across surrogate models were minimal. Overall, our findings suggest that smaller-parameter models generally serve as more effective surrogates for robust texture transfer across models.

it to the pre-trained voxel grid $V^{(\text{rgb})(\text{f})}$.

During training, we randomly select a viewpoint $v$ to render the corresponding 2D image $I_v$. To improve resilience to unknown corruptions, following the experimental setup in (Hendrycks & Dietterich, 2018), we introduce random types of corruption to the image $I_v$. Here, we apply 15 corruption types, including gaussian noise, shot noise, impulse noise, glass blur, defocus blur, zoom blur, motion blur, fog, frost, snow, contrast, brightness, JPEG compression, pixelation, and elastic transformation. The details of these corruptions can be found in supplementary A. This phase is similar to data augmentation in standard training.

$$I'_v \leftarrow C(I_v(V^{(\text{density})(\text{f})}, V^{(\text{rgb})(\text{f})} + \delta), s), \parallel \delta \parallel < \epsilon \quad (2)$$

where $s$ represent severity level and $\epsilon$ is the bound of $\delta$. There are five different severity levels for each type corruption, which will be selected randomly in training. This approach aims to prevent the model's recognition accuracy of objects under normal imaging conditions from reducing when only using high-severity corruptions. The degraded image $I'_v$ can be viewed as low-quality data obtained in complex real-world environments. We then use a proxy classifier $f_\theta$ to make predictions on $I'_v$ and compute the cross-entropy

loss between the output and the true label $y$.

$$\delta \leftarrow \nabla_\delta \mathcal{L}_{ce}(f_\theta(I'_v), y) \quad (3)$$

Through back-propagation, perturbation $\delta$ can be optimized to make the object more robust in any random view direction and any corruption.

### 3.4. Universal Robust Texture

Optimizing the surface texture of a single object to make its 2D images more robust has been explored in previous research (Salman et al., 2021). In comparison, we propose a more lightweight and flexible framework that enhances optimization speed by nearly 50 times. Leveraging the flexibility of our framework, we further investigate universal robust textures (URT). For a set of objects $\mathcal{V}$ belonging to category $y$ but differing in shape and appearance, we reconstruct each object's voxel grid $V_i^{(\text{density})(\text{f})}$ and $V_i^{(\text{rgb})(\text{f})}$ using multi-view images, with all grids at the same resolution. As in 3.3, we initialize a perturbation $\delta_U$ with the same shape as $V_i^{(\text{rgb})(\text{f})}$ at the start of training. Then, in each iteration, we randomly select a pair of voxel grids from the set and render a 2D image $I_v$ from a random viewpoint. Similarly, we apply corruptions of random types and inten-

sities to obtain $I'_v$, which is then passed to a downstream model for prediction.

$$I'_v \leftarrow C(I_v(V_i^{(\text{density})(\text{f})}, V^{(\text{rgb})(\text{f})} + \delta_U), s), V_i \in \mathcal{V} \quad (4)$$

At the end of training, we obtain a robust universal texture that adapts to different $V^{(\text{density})(\text{f})}$ objects within the same category and generalizes successfully to unseen objects.

### 3.5. Transfering Performance With Proxy Model

During training, a surrogate model is required to obtain gradient information. However, in practical scenarios, access to the internal parameters of the target model may not be feasible. Thus, whether the robust texture generated using a surrogate model can effectively transfer to an unknown model remains to be validated. Fortunately, our findings confirm that robust textures exhibit properties similar to adversarial perturbations, in that they can be transferred across models. This observation leads us to consider the relationship between the choice of surrogate model and transferability performance.

It is well known that adversarial perturbations generated by a model with higher generalization ability tend to be more transferable. Intuitively, if an adversarial example can deceive a stronger model, it is more likely to deceive a weaker model as well. In our task, we hypothesize the reverse relationship: if a robust example enables a weaker model to recognize it correctly, it is more likely to succeed with a stronger model. To substantiate this hypothesis, we conducted a set of comparative experiments. Using ResNet models with the same architecture but different parameter sizes—specifically, ResNet-18, ResNet-34, ResNet-50, ResNet-101, and ResNet-152—we generated robust textures for 10 randomly selected objects with each variant as a surrogate model, then transferred the textures to other ResNet models for evaluation. From Fig. 3, we observe that, aside from textures generated using the model itself as the surrogate in a white-box training setting, textures created with ResNet-18 as the surrogate model perform well across other models, particularly those with initially poorer performance, such as ResNet-34 and ResNet-50. For ResNet-101 and ResNet-152, however, due to their relatively strong initial performance, the differences among textures generated by various surrogate models are minimal when transferred to these models. Taking these findings together, we conclude that choosing a surrogate model with weaker performance can better accommodate a broader range of transfer scenarios.

## 4. Experiment

In this section, we conduct extensive experiments to demonstrate the superiority of our proposed method.

**Dataset.** Since most classification models are trained on the ImageNet dataset, and typical NeRF datasets lack category labels, we used the IM3D (Ruan et al., 2023) dataset to evaluate our method. This dataset includes 40 classes from ImageNet, with each class containing 10 objects, and each object represented by 100 rendered images from hemispherical viewpoints. When optimizing a single robust texture, we utilized random viewpoints for texture optimization and validated with sampled viewpoints from the dataset. For optimizing a generic robust texture, we randomly selected 8 objects from each category for training, while the remaining 2 objects were used for validation and testing, respectively.

**Testing Models.** To evaluate classification performance, we selected ResNet-18 (He et al., 2016) and VGG16 (Simonyan, 2014) as proxy models during the training process for each method. Additionally, since our approach enhances robustness from a data-centric perspective, the augmented data remain effective across various classification models. Therefore, we further selected ResNet-50, ResNet-152, MobileNetV2 (Sandler et al., 2018), Inception-V3 (Szegedy et al., 2016), ViT-b-16 (Dosovitskiy et al., 2020), and Swin-Small (Liu et al., 2021) as transfer models to assess performance on these architectures.

**Metrics.** We employed two metrics to assess model performance on corrupted images: accuracy and Corruption Error (CE). Accuracy, a commonly used metric in classification tasks, is defined as the proportion of correctly classified samples over the total number of samples. Additionally, to evaluate model robustness against corruption, the Corruption Error (CE) proposed by (Hendrycks & Dietterich, 2018) quantifies the performance degradation before and after applying corruption. Specific calculation methods are provided in the supplementary B. In this study, we use Relative mCE alongside accuracy as evaluation metrics.

**Implementation Details.** Our experiments were conducted on an Nvidia H100 GPU. The voxel grid resolution for 3D reconstruction was set to $N_x = N_y = N_z = 160$, with the training iterations set to 2000. The Bound of $\delta$ $\epsilon = 5.0$. During the texture optimization phase, the number of training iterations was set to 8000.

### 4.1. Performance on Proxy Model

We first analyze the test results on the surrogate model. As shown in Tab 1, we report accuracy under different fixed and random corruption intensities, as well as the Corruption Error (CE) for various types of corruptions. From the table, it can be observed that although performing well on ImageNet data, unfortunately, the methods fine-tuned on multi-view images exhibit a poor performance in our task setup. Among these methods, both DCP (Liu et al., 2024) and Unadv (Salman et al., 2021) damaged the robustness of original model. We think the architecture of DCP may

*Table 1.* Testing performance on proxy model. We report classification accuracy under conditions of no corruption (none), severity 1 through 5, and random severity, along with the mean corruption error (mCE) and relative mean corruption error (R.mCE) across 15 types of corruption. The top-performing results are represented in **bold**.

| PROXY | METHOD | TYPE | ACCURACY↑ | | | | | | | CE↓ | |
|---|---|---|---|---|---|---|---|---|---|---|---|
| | | | NONE | 1 | 2 | 3 | 4 | 5 | RANDOM | MCE | R.mCE |
| VGG16 | CLEAN | - | 0.3785 | 0.3088 | 0.2611 | 0.2360 | 0.1958 | 0.1496 | 0.2303 | - | - |
| | URIE (SON ET AL., 2020) | RESTORATION | 0.5645 | 0.5217 | 0.4898 | 0.4716 | 0.4396 | 0.4027 | 0.4531 | 0.6698 | 0.6702 |
| | VQSA (YANG ET AL., 2023) | FINETUNE | 0.8917 | 0.8259 | 0.7945 | 0.7521 | 0.7213 | 0.6847 | 0.7406 | 0.3020 | 0.3455 |
| | DCP (LIU ET AL., 2024) | FINETUNE | 0.1817 | 0.1377 | 0.1106 | 0.0972 | 0.0803 | 0.0614 | 0.0967 | 1.7976 | 1.8520 |
| | UNADV (SALMAN ET AL., 2021) | AUGMENTATION | 0.3592 | 0.3352 | 0.3126 | 0.2927 | 0.2708 | 0.2354 | 0.3181 | 1.0064 | 1.0265 |
| | OURS (SINGLE OBJ.) | AUGMENTATION | **0.9270** | **0.9114** | **0.8874** | **0.8628** | **0.8175** | **0.7772** | **0.8540** | **0.1746** | **0.1878** |
| | OURS (UNIVERSAL) | AUGMENTATION | 0.4150 | 0.3560 | 0.3245 | 0.3112 | 0.2825 | 0.2555 | 0.3247 | 0.9831 | 1.0253 |
| RESNET-18 | CLEAN | - | 0.3645 | 0.3064 | 0.2670 | 0.2391 | 0.2044 | 0.1656 | 0.2365 | - | - |
| | URIE (SON ET AL., 2020) | RESTORATION | 0.5773 | 0.5300 | 0.4997 | 0.4820 | 0.4530 | 0.4156 | 0.4713 | 0.6586 | 0.6636 |
| | VQSA (YANG ET AL., 2023) | FINETUNE | 0.8834 | 0.8436 | 0.8220 | 0.7952 | 0.7519 | 0.7262 | 0.8051 | 0.3125 | 0.3332 |
| | DCP (LIU ET AL., 2024) | FINETUNE | 0.3084 | 0.1853 | 0.1586 | 0.1311 | 0.1108 | 0.0748 | 0.1246 | 1.8025 | 1.8262 |
| | UNADV (SALMAN ET AL., 2021) | AUGMENTATION | 0.3132 | 0.2715 | 0.2641 | 0.2458 | 0.2039 | 0.1849 | 0.2413 | 1.1716 | 1.2202 |
| | OURS (SINGLE OBJ.) | AUGMENTATION | **0.9205** | **0.9073** | **0.8846** | **0.8590** | **0.8126** | **0.7769** | **0.8427** | **0.1771** | **0.1901** |
| | OURS (UNIVERSAL) | AUGMENTATION | 0.3850 | 0.3377 | 0.3097 | 0.2942 | 0.2748 | 0.2568 | 0.3178 | 1.0490 | 1.0408 |

*Table 2.* Transferable performance on other models. We report the average classification accuracy (Ave.Acc) under severity 1 through 5, along with the relative mean corruption error (R.mCE) across 15 types of corruption. The top-performing results are represented in **bold**.

| PROXY | METHOD | RESNET-50 | | RESNET-152 | | MOBILENETV2 | | INCEPTION-V3 | | VIT-B-16 | | SWIN-SMALL | |
|---|---|---|---|---|---|---|---|---|---|---|---|---|---|
| | | AVE.ACC | R.MCE | AVE.ACC | R.MCE | AVE.ACC | R.MCE | AVE.ACC | R.MCE | AVE.ACC | R.MCE | AVE.ACC | R.MCE |
| VGG16 | URIE (SON ET AL., 2020) | 0.3241 | 1.6237 | 0.3728 | 1.5716 | 0.2835 | 0.9320 | 0.2959 | 1.5178 | 0.3215 | 1.4302 | 0.4252 | 1.1846 |
| | UNADV (SALMAN ET AL., 2021) | 0.2745 | 2.2163 | 0.3271 | 1.8688 | 0.2103 | 1.3102 | 0.2642 | 1.7422 | 0.3704 | 1.2525 | 0.3652 | 1.8482 |
| | OURS (SINGLE OBJ.) | **0.6019** | **0.6477** | **0.6306** | **0.6365** | **0.4749** | **0.7040** | **0.5726** | **0.6435** | **0.5449** | **0.7177** | **0.6175** | **0.7034** |
| | OURS (UNIVERSAL) | 0.3366 | 1.5723 | 0.3720 | 1.6258 | 0.2405 | 1.1922 | 0.3112 | 1.2648 | 0.3596 | 1.3477 | 0.3618 | 1.8525 |
| RESNET-18 | URIE (SON ET AL., 2020) | 0.3472 | 1.6995 | 0.3971 | 1.5741 | 0.3187 | 0.9157 | 0.3115 | 1.2507 | 0.3487 | 1.4351 | 0.4127 | 1.0458 |
| | UNADV (SALMAN ET AL., 2021) | 0.3127 | 1.0964 | 0.3281 | 1.0348 | 0.2658 | 1.0823 | 0.2694 | 1.5120 | 0.3577 | 1.3832 | 0.3901 | 1.3538 |
| | OURS (SINGLE OBJ.) | **0.6684** | **0.5160** | **0.6823** | **0.5449** | **0.4631** | **0.6007** | **0.5221** | **0.5625** | **0.6028** | **0.6273** | **0.6612** | **0.6000** |
| | OURS (UNIVERSAL) | 0.3481 | 1.6376 | 0.3967 | 1.5970 | 0.2457 | 1.1883 | 0.3220 | 1.2413 | 0.3600 | 1.3711 | 0.3647 | 1.8556 |

only be suitable for natural images like ImageNet, while the Unadv method shows large performance variations across objects of different geometries, with average performance still lower than that of the original data. URIE (Son et al., 2020) partly enhanced the robustness of original model by removing corruptions but its performance is still unsatisfying. Although VQSA (Yang et al., 2023) performs well, it requires a long time to fine-tune the model, and this fine-tuning time increases exponentially with the amount of data, making it difficult to meet the demands of practical applications. In contrast, our proposed method not only demonstrates strong generalization across corruption intensities but also significantly improves the accuracy of the original model when recognizing uncorrupted images. As for the universal robust texture, although it did not outperform the texture optimized for a single object, it offers higher practical applicability and demonstrates superior performance compared to most baseline methods.

## 4.2. Transferable Performance Cross Various Models

Since our proposed method does not rely on fine-tuning a specific classification model, the generated robust textures can be transferred for use on other models. We further tested the performance of textures generated with different surrogate models when transferred to models of various architectures. For comparison, we selected URIE and Unadv, both of which are transferable, as baseline methods. We used average accuracy and relative mean corruption error as evaluation metrics. As shown in Tab 2, when transferred to other models, textures trained with ResNet-18 performed slightly better than those based on VGG16, suggesting that the features learned by ResNet-18 may be more similar to those in the transfer models. Additionally, because models like ResNet-152, ViT-b-16, and Swin-small already exhibit good robustness against corruptions, most methods had a counterproductive effect on these models, even leading to worse performance. We observed that our method consistently outperformed others when transferred across various models. In contrast, URIE and Unadv tend to degrade the natural feature representations learned by the models, thereby reducing accuracy. Comparing individually optimized textures for each object with universal robust textures shows that the former have superior cross-model generalization performance. This finding suggests that further enhancement of generalization may be necessary to develop highly transferable universal robust textures.

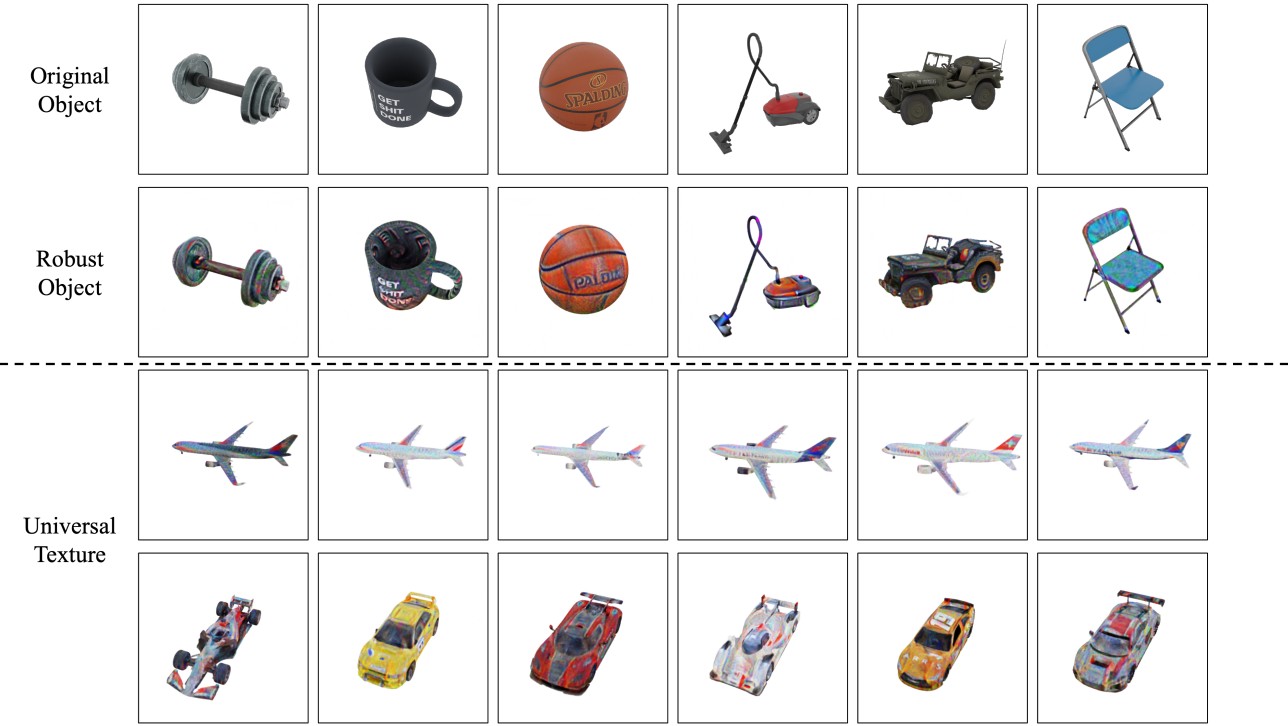

*Figure 4.* A visual comparison between the original appearance of various objects and their appearance after adding the robust texture is presented. It can be observed that the optimized robust texture does not significantly alter the objects' original appearance, thus preserving normal human perception. However, these subtle modifications to the appearance grant the objects considerable robustness under low-quality imaging conditions. In cases where low-quality images of the original appearance lead to misclassification by downstream models, objects with our optimized texture still enable stable and accurate predictions by the classifier.

### 4.3. Ablation Study

In this section, we examine the impact of different hyperparameter configurations on the final performance. In our proposed method, the primary hyperparameters are the voxel grid resolution and the boundary $\epsilon$ of the generated perturbation. We evaluated the final performance of the textures obtained with voxel grid resolutions of 160 and 320 and with $\epsilon$ values of 1.0 and 5.0. As shown in Tab 3, regardless of the model used as the proxy, results with $\epsilon = 5.0$ consistently outperform those with $\epsilon = 1.0$, indicating that a larger boundary value generally leads to better performance. For voxel grid resolution, when using VGG16 as the proxy model, a denser voxel grid with $\epsilon = 5.0$ slightly decreases the final performance. Conversely, for ResNet-18, increasing the voxel grid resolution consistently improves results. We hypothesize that the increased voxel grid resolution may lead to overfitting on smaller models, thereby impacting generalization.

*Table 3.* Performance comparison under different hyperparameter combinations. We report the average classification accuracy (Ave.Acc) under severity 1 through 5, along with the relative mean corruption error (R.mCE) across 15 types of corruption.

| Proxy | Grid | $\epsilon$ | Ave.Acc | R.mCE |
|---|---|---|---|---|
| | 160 | 1.0 | 0.6857 | 0.3203 |
| VGG16 | 160 | 5.0 | 0.8513 | 0.1878 |
| | 320 | 1.0 | 0.7063 | 0.2985 |
| | 320 | 5.0 | 0.8492 | 0.1958 |
| | 160 | 1.0 | 0.6531 | 0.3478 |
| ResNet-18 | 160 | 5.0 | 0.8481 | 0.1901 |
| | 320 | 1.0 | 0.6762 | 0.3511 |
| | 320 | 5.0 | 0.8504 | 0.1837 |

## 5. Conclusions

In this paper, we propose a data-driven approach to enhance robustness against low-quality imaging. Our method reconstructs a given object in 3D by sampling multi-view images to obtain its voxel representation, then optimizes a perturbation on its color grid to alter its appearance, thereby achieving robustness against various types and intensities

of corruption. Additionally, we introduce a universal robust texture, which optimizes the appearance of multiple objects with different geometries within the same category to obtain a transferable texture that generalizes to zero-shot objects. We further analyze the performance of textures obtained using different proxy models, summarizing the influence of the proxy model on the final outcome. Extensive experiments demonstrate the effectiveness of our proposed method, which outperforms existing image restoration and model fine-tuning approaches across different experimental settings, and retains effectiveness when transferred to models with different architectures.

## Acknowledgment

This research was supported in part by JSPS KAKENHI Grant Numbers 24KK0209, 24K22318, 22H00529, and JST-Mirai Program JPMJMI23G1.

## Impact Statement

This paper presents work whose goal is to advance the field of Machine Learning. There are many potential societal consequences of our work, none which we feel must be specifically highlighted here.

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

# A. Details of 15 Types of Corruptions

In this section, we provide the detailed calculation methods for each type of corruption. Given an input tensor $\mathbf{X} \in \mathbb{R}^{B \times 3 \times H \times W}$, let $\mathbf{Y}$ denote the tensor of corrupted low-quality images, then for

## A.1. Gaussian Noise:

$$\mathbf{Y} = \text{clamp}\Big(\mathbf{X} + \mathbf{Z}, \, 0, 1\Big), \quad \mathbf{Z} \sim \mathcal{N}(0, \sigma^2 \mathbf{I}), \quad (5)$$

where $\mathbf{Z} \in \mathbb{R}^{B \times 3 \times H \times W}$. For severity ranging from 1 to 5, the value of $\sigma$ is set to 0.08, 0.12, 0.18, 0.26, and 0.38, respectively.

## A.2. Shot Noise:

$$\mathbf{Y} = \text{clamp}\Big(\frac{\mathbf{P}}{c}, \, 0, 1\Big), \quad \mathbf{P} \sim Poisson(c \cdot \mathbf{X}). \quad (6)$$

For severity ranging from 1 to 5, the value of scaling factor $c$ is set to 60, 25, 12, 5 and 3, respectively.

## A.3. Impulse Noise:

$$\mathbf{Y} = \text{clamp}\Big(\mathbf{S} \circ \mathbf{B}(1 - \mathbf{S}) \circ \mathbf{X}, 0, 1\Big), \quad (7)$$

where $\mathbf{S} \sim Bernoulli(r)$ and $\mathbf{B} \sim Bernoulli(0.5)$. For severity ranging from 1 to 5, the value of $r$ 0.03, 0.06, 0.09, 0.17 and 0.27, respectively.

## A.4. Glass Blur:

To explain Glass blur, it is necessary first to introduce **Gaussian blur**. A **Gaussian kernel** $G(u, v)$ with a standard deviation $\sigma$ and size $k$ is defined as:

$$G(u, v) = \frac{1}{2\pi\sigma^2} \exp\Big(-\frac{u^2 + v^2}{2\sigma^2}\Big) \quad (8)$$

where $u, v \in -\lfloor k/2 \rfloor, ..., \lfloor k/2 \rfloor$ is the coordinate index of the gaussian kernel. The values in the kernel will be normalized so that:

$$\sum_{u,v} G(u, v) = 1 \quad (9)$$

The output tensor $\mathbf{Y}$ of gaussian blur is calculated by conducting 2D convolution on the input tensor $\mathbf{X}$ and gaussian kernel $G$:

$$\mathbf{Y} = \mathbf{X} * G \quad (10)$$

Glass blur is achieved by combining gaussian blur and random pixel swap. Given an input tensor $\mathbf{X} \in \mathbb{R}^{B \times 3 \times H \times W}$,

we first conduct gaussian blur to it to get a blurred version $\mathbf{X}'$. Then a set of pixel $\mathcal{C}$ is selected according to:

$$\mathcal{C} = \{(b, x, y) | b \in [1, B], x \in [r, W - r], y \in [r, H - r]\} \quad (11)$$

where $b$ is the index in batch and $(x, y)$ is pixel coordinate. $r$ is the radius for selecting pixels. Then we do $T$ times pixel value swaps. For each trial $t = 1, 2, ..., T$, generate random offset for every $(b, x, y) \in \mathcal{C}$:

$$(\Delta x, \Delta y) \sim \text{Uniform}(-r, r) \quad (12)$$

the new coordinate is:

$$(x', y') = (x + \Delta x, y + \Delta y) \quad (13)$$

then swap pixel values for new coordinates and old coordinates:

$$\mathbf{X}'[b, :, x', y'] \leftrightarrow \mathbf{X}'[b, :, x, y] \quad (14)$$

After $T$ times pixel value swaps we can get a corrupted tensor $\mathbf{X}''$. Finally do gaussian blur to $\mathbf{X}''$ again and the output can be seen as the glass blur version of $\mathbf{X}$. For severity ranging from 1 to 5, the values of gaussian kernel's standard deviation $\sigma$, the radius $r$ and trial number $T$ are (0.7, 1, 2), (0.9, 2, 1), (1, 2, 3), (1.1, 3, 2) and (1.5, 4, 2), respectively.

## A.5. Defocus Blur

Given a radius $r$, we establish an initial grid coordinate system $\mathcal{G}$ first:

$$\mathcal{G} = \{(x, y) | x \in [-R, R], y \in [-R, R]\} \quad (15)$$

where $R = \max(8, r)$. Then we construct a circular blur kernel $K$ based on this grid coordinate system:

$$K(x, y) = \begin{cases} 1 & \text{if } x^2 + y^2 \le r^2, \\ 0 & \text{otherwise} \end{cases} \quad (16)$$

The kernel should be normalized so that $\sum_{(x,y)} K(x, y) = 1$. Then conduct gaussian blur on this kernel to get an anti-aliasing smoothing kernel $K'$. Then do 2D convolution on the input tensor by using blur kenel $K'$:

$$\mathbf{Y} = \mathbf{X} * K' \quad (17)$$

For severity ranging from 1 to 5, the radius $r$ and standard deviation $\sigma$ are set to (3, 0.1), (4, 0.5), (6, 0.5), (8, 0.5) and (10, 0.5), respectively.

## A.6. Motion Blur

Given an offset angle $\theta_{\text{offset}}$, we first generate a random angle $\theta = \theta_{\text{offset}} + \text{Uniform}(-45°, 45°)$ for input tensor $\mathbf{X}$. Then calculate the blur offset in horizontal and vertical direction:

$$\text{clip}_h = \lfloor r_k \cdot \cos(\theta) \rceil, \quad \text{clip}_v = \lfloor r_k \cdot \sin(\theta) \rceil \quad (18)$$

A motion blur kernel $K \in \mathbb{R}^{(2r_k+1)\times(2r_k+1)}$ with radius $r_k$ and standard deviation $\sigma_k$ can be created by stretching a gaussian kernel in direction $\theta$. The motion blurred output $\mathbf{Y}$ can be obtained by doing 2D convolution on input tensor $\mathbf{X}$ using $K$ and clamping:

$$Y = \text{clamp}(K * \mathbf{X}, \, 0, \, 1) \quad (19)$$

In our setting the $\theta_{\text{offset}}$ is set to 0. For severity ranging from 1 to 5, the values of $r_k$ and $\sigma_k$ are set to (10, 3), (15, 5), (15, 8), (15, 12) and (20, 15), respectively.

## A.7. Zoom Blur

Given the number of scaling factors $K$, we first generate a set of scaling factors $z_k$:

$$\{z_k\}_{k=1}^K, \quad z_k = 1 + k \cdot s, \quad z_k < \max \text{zoom} \quad (20)$$

where $s$ is the size of zoom step. For each scaling factor $z_k$, rescale the input tensor $\mathbf{X}$ so that $\mathbf{X}_k = \text{Scale}(\mathbf{X}, z_k)$. Then crop the central region with the same size of $\mathbf{X}$ from the rescaled tensor $\mathbf{X}_k$:

$$\mathbf{X}_k^{trim} = \mathbf{X}_k[:,:, \Delta h : \Delta h + H, \Delta w : \Delta w + W], \quad (21)$$

where

$$\Delta h = \Delta w = \frac{\text{size}(\mathbf{X_k}) - \text{size}(\mathbf{X})}{2} \quad (22)$$

The output $\mathbf{Y}$ is calculated by:

$$\mathbf{Y} = \text{clamp}\left(\frac{\mathbf{X} + \sum_{k=1}^K \mathbf{X}_k^{trim}}{K+1}, 0, 1\right) \quad (23)$$

For severity ranging from 1 to 5, the values of max zoom and zoom step $s$ are set to (1.11, 0.01), (1.16, 0.01), (1.21, 0.02), (1.26, 0.02) and (1.31, 0.03), respectively.

## A.8. Fog

To add fog corruption on the original image, we should first generate a heightmap by using **Diamond-Square** algorithm. Given a mapsize and a factor $wd$ that controlling random wave decay, we initialize a square grid $M \in \mathbb{R}^{mapsize \times mapsize}$ with $M[0,0] = 0$. First do **square step**:

$$M\left[i+\tfrac{s}{2}, j+\tfrac{s}{2}\right] = \tfrac{M[i,j]+M[i+s,j]+M[i,j+s]+M[i+s,j+s]}{4} + \text{wibble}(s) \quad (24)$$

where $\text{wibble}(s) \sim \mathcal{U}(-\text{wibble}, \text{wibble})$. The value of wibble is set to 100 at the beginning and decreases with step size $s$. Then do **diamond step**:

$$M[i, j+\tfrac{s}{2}] = \tfrac{M[i,j]+M[i,j+s]+M[i-\tfrac{s}{2},j+\tfrac{s}{2}]+M[i+\tfrac{s}{2},j+\tfrac{s}{2}]}{4} + \text{wibble}(s) \quad (25)$$

After each Square and Diamond operation, the step size $s$ is halved, and the the amplitude of random wave is reduced

wibble $\leftarrow$ wibble$/wd$. Finally the heightmap is normalized to $[0, 1]$ and output. We defined this whole process as $diamond\_square$. For an input tensor $\mathbf{X}$, we can generate fractal noise for it:

$$F = diamond\_square(\mathbf{X}, mapsize, wd) \quad (26)$$

where the $mapsize = 2^{\lceil \log_2(\max(H,W)) \rceil}$. Then add the fractal noise $F$ to input tensor $\mathbf{X}$ to get $\mathbf{X}' = \mathbf{X} + \text{fog\_mixin} \cdot F$. Finally do normalization and clamp:

$$\mathbf{Y} = \text{clamp}\left(\mathbf{X}' \cdot \frac{\max(\mathbf{X}')}{\max(\mathbf{X}') + \text{fog\_mixin}}, 0, 1\right) \quad (27)$$

For severity ranging from 1 to 5, the values of fog\_mixin and wibble decay $wd$ are set to (1.6, 2), (2.1, 2), (2.6, 1.7), (2.5, 1.5) and (3., 1.4), respectively.

## A.9. Frost

To add frost corruption to the original input tensor, we randomly selected $B$ samples from a set that containing several frost images and add them with $\mathbf{X}$ directly:

$$\mathbf{Y} = \text{clamp}(c_i \cdot \mathbf{X} + c_f \cdot F, 0, 1) \quad (28)$$

where $c_i$, $c_f$ and $F$ represent the coefficient of input, the coefficient of frost images and the batch of frost images, respectively. For severity ranging from 1 to 5, the values of $c_i$ and $c_f$ are set to (1, 0.4), (0.8, 0.6), (0.7, 0.7), (0.65, 0.7) and (0.6, 0.75), respectively.

## A.10. Snow

To add snow corruption on an input tensor, we first randomly generate a noise layer:

$$S = \mathcal{N}(\mu, \sigma), S \in \mathbb{R}^{B \times 1 \times H \times W} \quad (29)$$

Then do rescale and crop to $S$:

$$S' = \text{Rescale}(S, zoom), S' \in \mathbb{R}^{B \times 1 \times H' \times W'}, \quad (30)$$

$$S_c = S'[:,:, t : t + H, t : t + W] \quad (31)$$

where $zoom$ is a zooming factor and $t = (H' - H)/2$ is the crop location. Then do threshold operation to $S_c$:

$$S_\tau(x) = \begin{cases} 0, & \text{if } S_c(x) < \tau, \\ S_c(x), & \text{otherwise} \end{cases} \quad (32)$$

Then apply **Motion Blur** to $S_\tau$ to get $S_{blur} = \text{MotionBlur}(S_\tau, r, s, \theta = -90°)$. For input tensor $\mathbf{X}$, convert it to gray scale image and do augmentation:

$$G = \text{GrayScale}(\mathbf{X}), G \in \mathbb{R}^{B \times 1 \times H \times W}, \quad (33)$$

$$G' = \max(\mathbf{X}, 1.5 \cdot G + 0.5) \quad (34)$$

Finally, mix up the augmented gray scale image and original image and add snow layer $S_{blur}$:

$$\mathbf{X}_{mix} = m \cdot \mathbf{X} + (1 - m) \cdot G' \tag{35}$$

$$\mathbf{Y} = \text{clamp}\Big(\mathbf{X}_{mix} + S_{blur} + \text{Rotate}(S_{blur}, 180°), 0, 1\Big) \tag{36}$$

For severity ranging from 1 to 5, the values of $\mu, \sigma, zoom$, $\tau, r, s$ and $m$ are set to (0.1, 0.3, 3, 0.5, 10, 4, 0.8), (0.2, 0.3, 2, 0.5, 12, 4, 0.7), (0.55, 0.3, 4, 0.9, 12, 8, 0.7), (0.55, 0.3, 4.5, 0.85, 12, 8, 0.65) and (0.55, 0.3, 2.5, 0.85, 12, 12, 0.55), respectively.

### A.11. Contrast

For each sample $X_b$ in input tensor $\mathbf{X}$, calculate the mean value $\mu_c$ for each channel:

$$\mu_c = \frac{1}{H \cdot W} \sum_{i=1}^{H} \sum_{j=1}^{W} X_{b,c,i,j} \tag{37}$$

And $\mathbf{Y}$ is the result of stretching or shrinking the pixel value based on the mean value:

$$\mathbf{Y} = \text{clamp}((\mathbf{X} - \mu) \cdot \alpha + \mu, 0, 1) \tag{38}$$

For severity ranging from 1 to 5, the value of $\mu$ is set to 0.4, 0.3, 0.2 0.1 and 0.05, respectively.

### A.12. Brightness

We first introduce two algorithm $rgb2hsv$ and $hsv2rgb$ that used for converting color space between RGB and HSV. Then the output $\mathbf{Y}$ with brightness corruption can be obtained by adding a perturbation to the $V$ channel:

$$\mathbf{X}_{hsv} = rgb2hsv(\mathbf{X}) \tag{39}$$

$$\mathbf{X}_{hsv}[:, 2, :, :] = \text{clamp}(\mathbf{X}_{hsv}[:, 2, :, :] + \delta, 0, 1) \tag{40}$$

$$\mathbf{Y} = \text{clamp}(hsv2rgb(\mathbf{X}_{hsv}), 0, 1) \tag{41}$$

For severity ranging from 1 to 5, the value of $\delta$ is set to 0.1, 0.2, 0.3, 0.4 and 0.5, respectively.

### A.13. JPEG Compression

We give an example algorithm to explain the process of JPEG compression.

### A.14. Pixelate

Given an input tensor $\mathbf{X}$, the pixelate corruption are achieved by reducing and enlarging $\mathbf{X}$:

$$\mathbf{X}_{reduced} = \text{Interpolate}(\mathbf{X}, (s \cdot H, s \cdot W), \text{mode} = \text{'bilinear'}) \tag{42}$$

---

**Algorithm 1** RGB to HSV Conversion

---

**Require:** $R, G, B \in [0, 1]$ ▷ Input RGB values
**Ensure:** $H \in [0, 360), S \in [0, 1], V \in [0, 1]$ ▷ Output HSV values
1: $C_{max} \leftarrow \max(R, G, B)$
2: $C_{min} \leftarrow \min(R, G, B)$
3: $\Delta \leftarrow C_{max} - C_{min}$
4: **if** $\Delta = 0$ **then**
5:     $H \leftarrow 0$
6: **else**
7:     **if** $C_{max} = R$ **then**
8:         $H \leftarrow 60 \cdot \frac{G-B}{\Delta} \mod 360$
9:     **else if** $C_{max} = G$ **then**
10:        $H \leftarrow 60 \cdot \frac{B-R}{\Delta} + 120$
11:     **else**
12:        $H \leftarrow 60 \cdot \frac{R-G}{\Delta} + 240$
13:     **end if**
14: **end if**
15: **if** $C_{max} = 0$ **then**
16:     $S \leftarrow 0$
17: **else**
18:     $S \leftarrow \frac{\Delta}{C_{max}}$
19: **end if**
20: $V \leftarrow C_{max}$ **return** $H, S, V$

---

**Algorithm 2** HSV to RGB Conversion

---

**Require:** $H \in [0, 360), S \in [0, 1], V \in [0, 1]$ ▷ Input HSV values
**Ensure:** $R, G, B \in [0, 1]$ ▷ Output RGB values
1: $C \leftarrow V \cdot S$
2: $X \leftarrow C \cdot (1 - |(H/60) \mod 2 - 1|)$
3: $m \leftarrow V - C$
4: **if** $0 \le H < 60$ **then**
5:     $(R', G', B') \leftarrow (C, X, 0)$
6: **else if** $60 \le H < 120$ **then**
7:     $(R', G', B') \leftarrow (X, C, 0)$
8: **else if** $120 \le H < 180$ **then**
9:     $(R', G', B') \leftarrow (0, C, X)$
10: **else if** $180 \le H < 240$ **then**
11:     $(R', G', B') \leftarrow (0, X, C)$
12: **else if** $240 \le H < 300$ **then**
13:     $(R', G', B') \leftarrow (X, 0, C)$
14: **else**
15:     $(R', G', B') \leftarrow (C, 0, X)$
16: **end if**
17: $R \leftarrow R' + m$
18: $G \leftarrow G' + m$
19: $B \leftarrow B' + m$ **return** $R, G, B$

---

$$\mathbf{Y} = \text{Interpolate}(\mathbf{X}_{reduced}, (H, W), \text{mode} = \text{'nearest'}) \tag{43}$$

---

**Algorithm 3** JPEG Compression Algorithm

---

**Require:** $\mathbf{X}$: Input tensor
**Ensure:** $C$: Compressed JPEG data
1: **Step 1: Convert to YCbCr (if RGB)**
2: **if** $\mathbf{X}$ is in RGB format **then**
3:     Convert $\mathbf{X}$ to YCbCr color space
4: **end if**
5: **Step 2: Divide image into 8x8 blocks**
6: Divide each channel of $\mathbf{X}$ into non-overlapping $8 \times 8$ blocks
7: **Step 3: Apply Discrete Cosine Transform (DCT)**
8: **for** each $8 \times 8$ block $B$ **do**
9:     Compute DCT coefficients $D \leftarrow \text{DCT}(B)$
10: **end for**
11: **Step 4: Quantize DCT coefficients**
12: **for** each $8 \times 8$ block $D$ **do**
13:     $Q \leftarrow D/Q_{table}$     ▷ Divide by quantization table
14:     Round $Q$ to nearest integer
15: **end for**
16: **Step 5: Encode the quantized coefficients**
17: **for** each block $Q$ **do**
18:     Perform zigzag scan to convert $Q$ to 1D array
19:     Apply Run-Length Encoding (RLE) to the zigzag array
20:     Use Huffman coding to encode the RLE data
21: **end for**
22: **Step 6: Combine encoded data**
23: $\mathbf{Y} \leftarrow$ Combine encoded data for all blocks with JPEG headers
24: **return** $\mathbf{Y}$

---

where $s$ is a scaling factor. For severity ranging from 1 to 5, the value of $s$ is set to 0.6, 0.5, 0.4, 0.32 and 0.29, respectively.

### A.15. Elastic Transform

We provide an example algorithm to explain the process of Elastic Transform. For severity ranging from 1 to 5, the values of parameters $c$ are set to (244 * 2, 244 * 0.7, 244 * 0.1), (244 * 2, 244 * 0.08, 244 * 0.2), (244 * 0.05, 244 * 0.01, 244 * 0.02), (244 * 0.07, 244 * 0.01, 244 * 0.02) and (244 * 0.12, 244 * 0.01, 244 * 0.02), respectively.

## B. Details of Validation Metrics

In our experiment, we employed two metrics to assess model performance on corrupted images: accuracy and Corruption Error (CE). Accuracy is defined as the proportion of true positive samples in all of samples:

$$Accuracy = \frac{TP}{TP + FP + TN + FN} \tag{44}$$

---

**Algorithm 4** Elastic Transformation

---

**Require:** Input image $\mathbf{X} \in \mathbb{R}^{B \times C \times H \times W}$, parameters $c = [c_0, c_1, c_2]$
**Ensure:** Transformed image $\mathbf{Y}$
1: Normalize the input image: $\mathbf{X}_{\text{norm}} \leftarrow \mathbf{X}/255$
2: Set shape shape $\leftarrow (B, C, H, W)$ and size shape_size $\leftarrow (H, W)$
3: **Step 1: Random Affine Transformation**
4: Compute center: $\mathbf{c}_{\text{center}} \leftarrow \left[\frac{H}{2}, \frac{W}{2}\right]$
5: Compute square size: $\mathbf{c}_{\text{square}} \leftarrow \frac{\min(H,W)}{3}$
6: Define reference points:

$$\mathbf{p}_1 \leftarrow \begin{bmatrix} \mathbf{c}_{\text{center}} + \mathbf{c}_{\text{square}} \\ \mathbf{c}_{\text{center}} + [\mathbf{c}_{\text{center},x} + \mathbf{c}_{\text{square}}, \mathbf{c}_{\text{center},y} - \mathbf{c}_{\text{square}}] \\ \mathbf{c}_{\text{center}} - \mathbf{c}_{\text{square}} \end{bmatrix}$$

7: Add random perturbation: $\mathbf{p}_2 \leftarrow \mathbf{p}_1 + \text{Uniform}(-c_2, c_2)$
8: Compute affine transform matrix: $\mathbf{M}_{\text{affine}} \leftarrow \text{getAffineTransform}(\mathbf{p}_1, \mathbf{p}_2)$
9: Apply affine transformation: $\mathbf{X}_{\text{affine}} \leftarrow \text{warpAffine}(\mathbf{X}_{\text{norm}}, \mathbf{M}_{\text{affine}})$
10: **Step 2: Generate Pixel Displacement Fields**
11: Compute random fields:

$$\Delta_x \leftarrow \text{Gaussian}(\text{Uniform}(-1, 1), c_1) \cdot c_0$$

$$\Delta_y \leftarrow \text{Gaussian}(\text{Uniform}(-1, 1), c_1) \cdot c_0$$

12: Reshape displacement fields: $\Delta_x, \Delta_y \in \mathbb{R}^{H \times W \times 1}$
13: **Step 3: Apply Coordinate Mapping**
14: Generate original grid: $(x, y, z) \leftarrow \text{meshgrid}([0, W], [0, H], [0, C])$
15: Add displacements: $x' \leftarrow x + \Delta_x, y' \leftarrow y + \Delta_y$
16: Map coordinates using interpolation:

$$\mathbf{X}_{\text{elastic}} \leftarrow \text{mapCoordinates}(\mathbf{X}_{\text{affine}}, (x', y', z), \text{order} = 1)$$

17: **Step 4: Normalize and Return**
18: Clip values: $\mathbf{Y} \leftarrow \text{clip}(\mathbf{X}_{\text{elastic}}, 0, 1) \cdot 255$
19: **return** $\mathbf{Y}$

---

For each of our experiments, since the images belong to the same category, so $TN = FN = 0$. We defined the accuracy under corruption $C$ and severity $i$ as $Accuracy_{C_i}$, thus the Average Accuracy (Ave.Acc) can be calculated by:

$$\text{Average Accuracy} = \sum_{i=1}^{5} \frac{Accuracy_{C_i}}{5} \tag{45}$$

Corruption Error (CE) is proposed to comprehensively evaluate a classifier's robustness to a given type of corruption. The first evaluation step is to take a trained classifier $f$, which has not been trained on IMAGENET-C, and com-

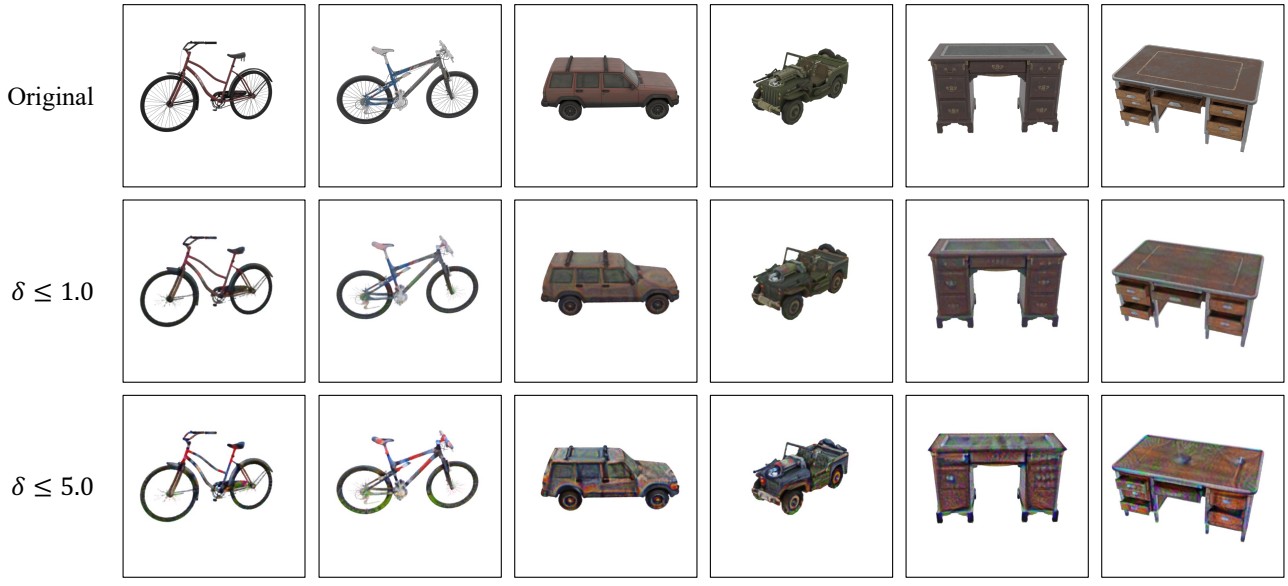

*Figure 5.* More visualization results of our robust texture.

pute the clean dataset top-1 error rate. Denote this error rate $E_{\text{clean}}^{f}$. The second step is to test the classifier on each corruption type $c$ at each level of severity $s$ ($1 \leq s \leq 5$). This top-1 error is written $E_{s,c}^{f}$. Before aggregating the classifier's performance across severities and corruption types, error rates should be made more comparable since different corruptions pose different levels of difficulty. We adjust for the varying difficulties by dividing by the errors of target classifier (when using VGG16 for testing, we use VGG16's errors). So Corruption Error is computed with the formula:

$$\text{CE}_{c}^{f} = \Big( \sum_{s=1}^{5} E_{s,c}^{f} \Big) / \Big( \sum_{s=1}^{5} E_{s,c}^{\text{target}} \Big) \tag{46}$$

The mean CE (or mCE for short) can be calculated by averaging the 15 Corruption Error values. The authors further proposed a metric to indicate the amount that a classifier declines on corrupted inputs, which named Relative Corruption Error:

$$\text{Relative CE}_{c}^{f} = \Big( \sum_{s=1}^{5} E_{s,c}^{f} - E_{\text{clean}}^{f} \Big) / \Big( \sum_{s=1}^{5} E_{s,c}^{\text{target}} - E_{\text{clean}}^{\text{target}} \Big) \tag{47}$$

Averaging these 15 Relative Corruption Errors results in the $Relative mCE$. This measures the relative robustness or the performance degradation when encountering corruption.

## C. More Visualization Results

We present additional visualization results to illustrate the differences between the textures generated by our method under varying $\delta$ boundaries and the original textures. The corresponding results are shown in Fig 5. All results were generated using ResNet-18 as the proxy model.

