# OpenReview forum: "You Always Recognize Me (YARM): Robust Texture Synthesis Against Multi-View Corruption"
_ICML.cc/2025/Conference — ICML 2025 poster_

### Official Review · Reviewer_mDnf · 2025-02-27

**Overall Recommendation:** 2

**Summary:**

This paper addresses real-world unknown image degradation that affects deep learning model performance. The authors propose a novel data-centric approach that optimizes the textures of 3D objects to enhance their robustness against corruption. The methodology is based on 3D voxel grid representations reconstructed from multi-view images. A classifier is used as a surrogate model to optimize textures, ensuring robustness against corruption during imaging.

## update after rebuttal

After I review the author's response and other reviewers' comments, I maintain my score. I appreciate the authors' efforts.

No higher score because the contribution doesn't reach the bar of ICML.

**Claims And Evidence:**

moderate, please see details in weaknesses.

**Essential References Not Discussed:**

see problems

**Experimental Designs Or Analyses:**

I think the experiment is not solid and enough.

**Methods And Evaluation Criteria:**

yes, they make sense in the proposed setting.

**Other Comments Or Suggestions:**

see problems

**Other Strengths And Weaknesses:**

Strengths:
1.Novel Data-Centric Perspective: Instead of modifying models or preprocessing images, object texture optimization is a new angle for improving robustness.
2.Transferability: While transferability is a core concern of the article, it indicates how the output of a single model can be generalized to other architectures.
3.Application of 3d reconstruction: Using voxel-based NeRF for texture optimization aligns well with modern 3D deep learning approaches.

Weaknesses:
1.Unpolished writing: line 162-163 and 199-200 is repetitive. Formula 3 is cross entropy loss but labeled as mse. Figure captions are
2.Lack of real-world experiments: The experiments are performed on synthetic datasets rather than real-world scenarios. But whether the method is applicable to real-world data is a key question mentioned in the intro
3.Scalability: 3d reconstruction makes this method difficult to scale up to larger images.

**Questions For Authors:**

1.The article discussed the existence of a universal robust texture. However the experiment results did not support the importance and necessity of this feature. Why do we need this additional computation if the single robust texture is already transferable? It is mentioned that hopefully this facilitates the classification of unseen data, but this is not verified.
2.Why does some texture generated from a stronger backbone like res152 lead to significant drop in performance for simpler architectures like res18 and res34?
3.How does the object become robust as the training progresses? Figure 4 is a bit confusing in displaying it.

**Relation To Broader Scientific Literature:**

see problems

**Theoretical Claims:**

limited theoretical analysis

---

> ### Author Rebuttal · Authors · 2025-03-31
>
> We greatly appreciate the reviewers’ thoughtful feedback and insightful suggestions. We have carefully reviewed all the comments and provide our detailed responses below.
>
> **A1.Regarding writing issues:**
> Thank you for pointing out these issues! We will correct these errors in the revised version of the paper.
>
> **A2.Regarding real-world experiments:**
> Thank you for your question! We fully understand the importance of real-world experiments. However, compared to previous methods (such as Unadv), we have already conducted extensive evaluations of our proposed method on a large dataset containing 40 categories and 400 object instances (whereas Unadv evaluated only 5 objects). We believe this is sufficient to demonstrate the generalization capability of our method. Nevertheless, to further address your concern, we have additionally conducted an experiment on a real-world scene. Specifically, we selected a real scene used in [1]. Since our method requires separating the foreground object, we manually segmented the foreground object from the multi-view images and performed 3D reconstruction and texture optimization accordingly. The experimental results based on ResNet-18 are presented in the table below. The visualization results can be found here https://p.sda1.dev/23/b6df0ef9d63f80fba42e36ce11994fed/real_scene.png
> |Method|none|1|2|3|4|5|random|m.CE|R.mCE|
> |:-:|:-:|:-:|:-:|:-:|:-:|:-:|:-:|:-:|:-:|
> |Clean|0.7155|0.7016|0.6373|0.5424|0.4390|0.3382|0.4959|-|-|
> |Ours|1.0000|0.9957|0.9786|0.8932|0.7009|0.5342|0.8417|0.2395|0.3720|
>
> **A3.Regarding scalability:**
> Thank you for your question! We believe that large-size images can be downscaled to fit our 3D reconstruction pipeline, and therefore, scalability to larger images is not a major concern in our method. In contrast, images with excessively low resolution may significantly degrade the quality of 3D reconstruction, which could in turn hinder the effectiveness of the optimized textures.
>
> **A4.Regarding universal robust textures:**
> Thank you for your question! Indeed, a single object-specific robust texture can only be transferred across different classifiers but cannot be directly transferred to other objects with different shapes, even within the same category. We proposed the concept of universal robust textures because object-specific texture synthesis is an inefficient process. Therefore, our goal is to generate a universal texture for each category, independent of the specific appearance and shape of individual instances. We have validated the feasibility of this idea through preliminary experiments. It is worth noting that the performance of universal robust textures reported in Tables 1 and 2 was evaluated on objects that were not used during training. This demonstrates that these textures indeed provide protection for previously unseen objects within the same category. However, the primary focus of this paper is on object-specific robust textures. As such, we did not devote significant effort to further improve the performance of category-level robust textures, which may explain why their current performance appears unsatisfactory. Nevertheless, we believe that the direction of category-level robust textures is worthy of further exploration. We have also discussed possible ways to improve their performance (see ***A6 to Reviewer rM6D***).
>
> **A5.Regarding selection of proxy model:**
> Please refer to ***A5 to Reviewer rM6D***.
>
> **A6.Regarding the question about Figure 4:**
> Thank you for your question! We apologize for the lack of clarity in the presentation of Figure 4. Specifically, the first row of Figure 4 shows the original appearance of the objects; the second row shows the appearance after applying object-specific robust texture optimization for each object; and the third row shows the appearance when applying the universal robust texture, generated for the "airplane" category, to objects of different shapes within the same category. As illustrated in Figure 1, we selected 8 objects from the "airplane" category as the training set. We reconstructed voxel grid representations for each of these 8 objects based on their multi-view images. Then, we initialized a random perturbation $\delta$ and, during each iteration of the optimization process, randomly selected one voxel grid to combine with $\delta$ and rendered the corresponding view images. In this way, the resulting category-level robust texture can be directly applied to unseen voxel grid representations of other objects belonging to the "airplane" category, enabling the rendering of corruption-robust views. We hope this explanation helps clarify your concerns regarding the universal robust texture and the content of Figure 4.

---

> > ### Comment · Reviewer_mDnf · 2025-04-01
> >
> > Given that my major concerns haven't been fully addressed, I insist on my initial score, recommending a weak reject to this paper.

---

> > > ### Author Response · Authors · 2025-04-02
> > >
> > > Dear Reviewer mDnf,
> > >
> > > We sincerely appreciate your response. If possible, we would be grateful if you could kindly specify which concerns you believe remain insufficiently addressed and, if feasible, offer constructive feedback that could help us further refine and strengthen our work.

---

### Official Review · Reviewer_eS8T · 2025-03-10

**Overall Recommendation:** 3

**Summary:**

This paper proposes a data-centric approach to enhance the performance of deep learning models in the presence of image degradation by utilizing multi-view 3D reconstruction and optimizing for a robust texture. They have shown that a generalized robust texture exists that can transfer across objects of the same category but with different geometries. From the experiments, they conclude that choosing a surrogate model with weaker performance can better accommodate a broader range of transfer scenarios. They have shown relevant experiments to show the performance of all the subparts. They have a dataset IM3D to show the effectiveness of their method. The dataset contains 40 different classes where each class contains 10 different objects and there are around 100 images for each object.

## Update after rebuttal
Authors have addressed my concerns. I keep my original rating 'weak accept'.

**Claims And Evidence:**

1. In Line 139-140, it is given that some previous methods have explored robustness in 3D objects. And the differences (they were not tested on large-scale datasets and other models, ...) are not convincing. Concepts look similar. Can you please specify what are the exact differences between your method and those methods? The difference is only where you are utilizing 3D reconstruction in your optimization?

2. Is it possible that, in Eq. (3), the optimized $\delta$ becomes zero, and the classifier predicts the correct class always?

**Essential References Not Discussed:**

NA

**Experimental Designs Or Analyses:**

Are the experiments fair? Can you please specify the training settings of the methods compared in Tabl 1 and 2? Have all methods been retrained/finetuned with the same dataset?

**Methods And Evaluation Criteria:**

Did you check the effectiveness of your method on some real data (real-world)? Other than the dataset IM3D you have used? I feel there should be some results on real data also.

**Other Comments Or Suggestions:**

NA

**Other Strengths And Weaknesses:**

Strengths are
1. The paper is well-written and it is interesting to read.
2. Proper experiments are shown.

Weaknesses
1. Fig. 1, 2, and 4 are not referred to in the texts and these figures are like some independent things in the paper. Readers may not see those figures if they are not cited in the text.
2. In Fig. 4, what is shown by universal texture? Not clear!
3. Line 209, column 2, why you have replaced the soft plus activation function with ReLU?
4. IM3D dataset is from which paper? Is it cited in the paper?
5. It is better to have more visual results of more different objects and also every method you have compared (if you can show what they have learned).

**Questions For Authors:**

Against each of the sections, questions are given. Please try to address my concerns to help me make a more favorable decision.

**Relation To Broader Scientific Literature:**

From the current version of the paper, I feel, the contribution of the paper is good to the scientific literature. But, please specify the exact differences from the previous literature as I have raised in one previous query.

**Theoretical Claims:**

No theoretical claims.

---

> ### Author Rebuttal · Authors · 2025-03-31
>
> We appreciate the reviewers’ thoughtful feedback and provide our detailed responses below.
>
> **A1.Regarding the specific differences from prior methods:**
> Thanks for your question! We acknowledge that our work is inspired by Unadv; however, there are clear differences that distinguish our approach from Unadv: 1.Optimization space: Unadv directly optimizes the pixel values of texture images in the 2D image space, while our method performs optimization in the voxel grid space. We believe the high dimensionality of the voxel grid allows for better optimization outcomes. 2. Visual appearance preservation: Directly optimizing texture images in pixel space often results in highly discrete pixel values, which can manifest as noise-like visual artifacts on the object surface. In contrast, the textures generated by our method exhibit a certain degree of local continuity. This advantage is particularly evident when applied to objects without initial textures (see ***A3***). 3.Efficiency: Our method is approximately 100 times faster than Unadv. Unadv's optimization is based on 3D mesh representations combined with a differentiable rendering framework, which typically requires 6 to 7 hours to optimize a single object. In comparison, our approach takes only 4 to 5 minutes to complete the optimization.
>
> **A2.Regarding Eq.3:**
> Thanks for your question! We acknowledge that such a situation is theoretically possible; however, the probability is very low. Specifically, for this to happen, the gradient of the $\delta$ term obtained through backpropagation during training would need to remain 0 throughout the entire process. Additionally, it would require that the multi-view images of the original object exhibit strong robustness against all types of corruptions. Furthermore, commonly used optimizers incorporate momentum terms, which help prevent the optimization from getting stuck in such a local optimum. Therefore, we believe this situation is unlikely to occur in practice.
>
> **A3.Regarding real-world data and other datasets:**
> The reason why we only used the IM3D dataset in our experiments is that our method requires datasets with specific characteristics: (1) multi-view images along with camera poses to enable 3D reconstruction; and (2) category labels aligned with ImageNet classes to support the classification task. Therefore, datasets commonly used in NeRF-related tasks are not directly applicable to our pipeline. Nevertheless, to demonstrate the generalization ability of our method, we selected 4 objects from the ModelNet40 dataset—originally designed for point cloud classification—and rendered multi-view images of them using Blender for evaluation. Experimental results based on VGG16 is shown below. Visualization results can be found here https://p.sda1.dev/23/b660fbea4c3a6d471effde89e4b4110f/modelnet.png. For real scene results please see ***A2 to Reviewer mDnf***.
> |Method|none|1|2|3|4|5|random|m.CE|R.mCE|
> |:-:|:-:|:-:|:-:|:-:|:-:|:-:|:-:|:-:|:-:|
> |Clean|0.7229|0.6296|0.6014|0.5693|0.5228|0.4596|0.5381|-|-|
> |URIE|0.3708|0.3235|0.3301|0.3067|0.2844|0.2326|0.3125|1.7991|1.9015|
> |VQSA|0.3854|0.4032|0.4081|0.3949|0.3838|0.3757|0.3851|1.4768|1.5398|
> |DCP|0.2819|0.2652|0.2138|0.1677|0.1429|0.1158|0.1443|2.3419|2.2930|
> |Unadv|0.1794|0.1620|0.1566|0.1283|0.1139|0.1031|0.1123|3.4739|4.6710|
> |Ours|0.9646|0.9334|0.8875|0.8599|0.7971|0.6982|0.8347|0.3463|0.3799|
>
> **A4.Regarding the fairness of the experiments:**
> Thanks for your question! For the baseline methods, we first evaluated their publicly released weights (most of which were pretrained on ImageNet) on the IM3D dataset. However, due to distribution differences, most of these methods exhibited suboptimal performance on IM3D. Therefore, we additionally fine-tuned URIE, VQSA, and DCP on the IM3D dataset to ensure a fair comparison. After fine-tuning, VQSA showed significant performance improvement, URIE exhibited a slight improvement, while DCP showed no noticeable improvement.
>
> **A5.Regarding Figure 4:**
> Thanks for your question! In Figure 4, we illustrate the appearance of universal robust texture when applied to multiple instances with different shapes within the same category (airplane). We will add further clarification to the caption of Figure 4 in the revised version and ensure that Figures 1, 2, and 4 are explicitly referenced in the main text.
>
> **A6.Regarding the use of ReLU:**
> This design choice was proposed by [1], where the authors claimed that replacing SoftPlus with ReLU better preserves the discontinuities present in real-world signals.
>
> **A7.Regarding the IM3D dataset:**
> The IM3D dataset is sourced from [2]. We apologize for the oversight of not citing it in the original submission. We will add the appropriate citation in the revised version.
>
> **A8.Regarding the visualization results:**
> Please refer to ***A3***.
>
> [1].ReLU Fields: The Little Non-linearity That Could.
> [2].Towards viewpoint-invariant visual recognition via adversarial training.

---

> > ### Comment · Reviewer_eS8T · 2025-04-04
> >
> > Authors have addressed my concerns. I keep my original rating 'weak accept'.

---

> > > ### Author Response · Authors · 2025-04-08
> > >
> > > Dear Reviewer eS8T,
> > >
> > > Thank you very much for acknowledging both the significance of our work and the efforts we have made in the rebuttal. If our response has adequately addressed your concerns, we would greatly appreciate it if you would consider raising your score. Your support would be very meaningful to us.
> > >
> > > Best regards,
> > >
> > > Authors of Paper 11436

---

### Official Review · Reviewer_rM6D · 2025-03-15

**Overall Recommendation:** 3

**Summary:**

The paper focuses on the reconstruction of a 3D object from a set of low-quality 2D images. The method applies 15 different corruption techniques during training to the respective images just like a data augmentation technique to make the classifier strong enough independent of the noise introduced. Furthermore, this research evaluates the classification performance of the reconstructed objects by transferring knowledge from the surrogate model to a larger architecture.

## update after rebuttal
I decided to keep my original score (weak accept).

**Claims And Evidence:**

The claims mentioned in the paper are supported by various studies that have explored similar approaches. The paper discusses the use of NeRF methods to justify the adoption of a voxel representation for generating the corresponding reconstruction.

Additionally, the method introduces universal adversarial perturbations (UAP) as a guiding mechanism for their universal robust texture, developed from a set of objects. The authors reference literature that examines how small perturbations can significantly affect classification outcomes.

However, the discussion on transfer learning in relation to the surrogate models is presented but not well supported by prior evidence. While it is well known that transfer learning can enhance the performance of a larger model by training a distilled version, the cited literature focuses on alternative approaches (segmentation models) that are not directly related to this specific method.

**Essential References Not Discussed:**

**Transfer Learning**
The paper by Gatys et al. (2016), "Image Style Transfer Using Convolutional Neural Networks," is an influential approach for transferring artistic styles between images. This work showed how CNNs previously optimized for object recognition could be repurposed to separate and recombine content and style representations of images. By defining content and style losses based on feature activations and feature correlations in different CNN layers, their method transfers the stylistic elements of artworks like Van Gogh's "The Starry Night," Munch's "The Scream," and Picasso's "Seated Nude" onto images.

**Image Restoration**
Several key image restoration papers are missing from the references. Zhang et al. (2017) showed how CNNs could effectively denoise images using residual learning. Dong et al. (2015) was the first major work applying deep learning to super-resolution. Isola et al. (2017) created pix2pix, which handles many image-to-image translation tasks including restoration. Wang et al. (2018) improved GAN-based restoration with ESRGAN, while Ulyanov et al. (2018) found that network architectures inherently contain useful priors for restoration. Including these works would give better context for comparing the paper's approach with restoration-based methods for handling corrupted images.

**Experimental Designs Or Analyses:**

The paper includes the ablation study on evaluating the performance of the textures obtained with different hyperparameter combinations and voxel grid resolutions.

The experimental design is valid for providing insights into the impact of these specific hyper-parameters, but a more comprehensive ablation study could have strengthened the paper's conclusions about optimal parameter settings across different scenarios. In addition, should be important to mention how many objects were run in these experiments and provide some figures where it is possible to visualize the differences.

**Methods And Evaluation Criteria:**

- The experiments for this method were conducted downstream using the IM3D dataset, which has been widely utilized in various applications, including mesh generation and image classification. However,  the method is not tested on other datasets, such as ShapeNet3D, Objaverse, or ModelNet40, to demonstrate its generalization to different objects.

- The paper does not provide visual results for other methods, only the respective metrics. In 3D reconstruction, it is particularly important to present at least one example from other approaches to allow for a proper comparison of the results.

**Other Comments Or Suggestions:**

N/A

**Other Strengths And Weaknesses:**

Strengths:
1. The key innovation is making object appearances inherently robust to corruption, rather than trying to repair corrupted images or complicate models. This directly benefits safety-critical systems like autonomous vehicles that must reliably identify objects in varying conditions.

2. This approach solves real-world problems where vision systems must function in harsh conditions. Unlike NeRF and Gaussian Splatting which assume perfect images, this method maintains performance despite weather effects, sensor noise, and motion blur.

Weaknesses:

1. **Missing ablation studies**: There's limited exploration of how different components of their pipeline contribute to the final performance, making it difficult to understand which aspects are most critical.

2. The performance seems highly dependent on the choice of proxy model during optimization, but the paper doesn't provide clear guidelines for selecting optimal proxy models for different scenarios. There are recent CNN's that can be study like ConvNext.

**Questions For Authors:**

1. the results in Figure 3 suggest that smaller models tend to be better surrogates for transferable texture optimization. Could you share insights into why this might be the case?

2. Table 1 shows a performance gap between universal robust textures and object-specific textures. Do you see pathways to improving universal texture performance, or are there fundamental limitations to what universal textures can achieve compared to object-specific ones?

3. Could your approach be extended to jointly optimize both textures and classification models in an end-to-end manner?

**Relation To Broader Scientific Literature:**

The paper leverages recent advances in neural radiance fields (NeRF) and their accelerated variants using voxel grid representations. The authors adapt these techniques for texture optimization rather than novel view synthesis. This paper is related to NeRF-based editing research, showing how voxel grid-based 3D representations can be used for specific downstream objectives.

**Theoretical Claims:**

All the theoretical claims are correct including the ones mentioned in the appendix about the formulas used for each corruption.

---

> ### Author Rebuttal · Authors · 2025-03-31
>
> We greatly appreciate the reviewers’ thoughtful feedback and insightful suggestions. We have carefully reviewed all the comments and provide our detailed responses below.
>
> **A1.Regarding experiments on ModelNet40 datasets:** Thanks for your question! Please refer to ***A3 to Reviewer eS8T***.
>
> **A2.Regarding visualization results:**
> Thanks for your question! Since VQSA and DCP are finetuning-based methods, we provided visualization results of URIE, Unadv and our method. Please refer to ***A3 to Reviewer eS8T*** for results on ModelNet40 and ***A2 to Reviewer mDnf*** for real scene.
>
> **A3.Regarding ablation study:**
> Thanks for your question! In fact, our method does not involve many detachable modules that allow for extensive ablation studies. Therefore, in the ablation study section of the paper, we primarily focused on analyzing the impact of voxel grid resolution and the boundary of texture perturbation on the final results. To further illustrate the superiority of voxel grids as a 3D representation in our framework, we additionally compared the performance of robust textures when using NeRF's MLP as the 3D representation. In this experiment, we selected one object from each category for evaluation. The results on Resnet18 is shown below:
> |Method|none|1|2|3|4|5|random|m.CE|R.mCE|
> |:-:|:-:|:-:|:-:|:-:|:-:|:-:|:-:|:-:|:-:|
> |Clean|0.3611|0.3289|0.2793|0.1855|0.1325|0.1156|0.1602|-|-|
> |MLP|0.4527|0.4078|0.3629|0.2774|0.2168|0.1825|0.2484|0.6873|0.7540|
> |Voxel(Ours)|0.8714|0.8598|0.7994|0.7182|0.6530|0.5796|0.6901|0.2617|0.2984|
>
> **A4.Regarding reference:**
> Thank you for your question! However, we would like to clarify that our work is not closely related to transfer learning or style transfer. The reason why the robust textures can be transferred across different classifiers is similar to the phenomenon observed in adversarial examples: different classifiers tend to learn feature spaces with certain similarities. Additionally, image restoration works have a different focus compared to ours. Traditional image restoration aims to recover as much visual detail of the original image as possible, whereas our objective is to enhance the performance of downstreaming models when encountering corrupted images. Image restoration is merely one possible approach that can be incorporated into our framework, but it is not the primary focus of our work.
>
> **A5.Regarding selection of proxy model:**
> Thanks for your question! As we explained in section 3.5, employing a proxy model with relatively weaker generalization ability can lead to better transferability to other model architectures. To further clarify this design choice, we consider our task as the reverse of an adversarial attack task. In prior work on adversarial attacks, it has been observed that when the proxy model itself has stronger generalization capability, the generated adversarial examples tend to exhibit higher transferability. Intuitively, this can be likened to a teacher-student scenario: if a question can challenge a very capable student, it is more likely to challenge a less capable one as well. In contrast, our objective is to enable the classifier to correctly recognize the input, which can be viewed as the opposite of adversarial attacks. In this analogy, it is akin to designing questions that students can answer correctly. If a question can be easily answered by a less capable student, it is naturally more likely to be correctly answered by a more capable one as well.
>
> **A6.Regarding universal robust textures:**
> Thanks for your question! We proposed universal robust textures because synthesizing robust textures for each individual instance is an inefficient process. However, since the main focus of this paper is on object-specific robust texture synthesis, we only conducted preliminary experiments to validate the feasibility of universal robust texture. We believe that the performance of universal robust textures can be further improved in the following ways: 1. A larger, category-organized multi-view 3D reconstruction dataset. In our current IM3D dataset, each category contains only 10 different objects, and we used only 8 of them to train the robust textures. This clearly limits their generalization capability. 2. More efficient 3D representation. In our current setting, we adopt voxel grids as the 3D representation. However, training universal robust textures on a larger set of objects inevitably requires increasing the capacity of voxel grids, which would significantly raise the GPU cost. NeRF-based representation may be better for scaling up category-level robust texture training.
>
> **A7.Regarding end-to-end training:**
> Thanks for your question! Our proposed data-centric approach for enhancing classifier performance is based on the assumption that, in industrial scenarios, deployed classifiers cannot be easily modified or retrained. But we think it's possible to jointly optimize both the textures and the classifier to get better performance.

---

> > ### Comment · Reviewer_rM6D · 2025-04-04
> >
> > Many thanks for the rebuttal. I decide to keep my acceptance score.

---

> > > ### Author Response · Authors · 2025-04-08
> > >
> > > Dear Reviewer rM6D,
> > >
> > > Thank you very much for acknowledging both the practical relevance of our work and the efforts we have made in the rebuttal. If our response has adequately addressed your concerns, we would greatly appreciate it if you would consider raising your score. Your support would be very meaningful to us.
> > >
> > > Best regards,
> > >
> > > Authors of Paper 11436.

---

### Decision · Program_Chairs · 2025-05-01

**Decision:**

Accept (poster)

**Comment:**

This paper received three reviews with mixed initial ratings (2 weak accepts, one weak reject). There was consensus around the importance of the problem considered -- reconstruction and inference from corrupted images -- a timely problem that can especially benefit safety-critical systems in robotics and other applications. There was also appreciation for the novelty of the proposed approach centered around the observation of a robust texture / appearance that is preserved across multiple views. The paper demonstrates the performance of the proposed approaches under a wide range of non-ideal conditions, including weather effects, sensor noise, and motion blur. This could have strong implications for real-world vision systems which must function robustly in challenging conditions.

There were some concerns raised around presentation, real-world evaluations and scalability, which were appropriately addressed in the authors' response. Given this, an accept decision is recommended.